# SARS-CoV-2 (COVID-19 pandemic) in Nigeria: Multi-institutional survey of knowledge, practices and perception amongst undergraduate veterinary medical students

Oluwawemimo Oluseun Adebowale[1]*, Olubukola Tolulope Adenubi[2], Hezekiah Kehinde Adesokan[3], Abimbola Adetokunbo Oloye[4], Noah Olumide Bankole[5], Oladotun Ebenezer Fadipe[6], Patience Oluwatoyin Ayo-Ajayi[7], Adebayo Koyuum Akinloye[8]

1 Department of Veterinary Public Health and Preventive Medicine, College of Veterinary Medicine, Federal University of Agriculture, Abeokuta, Ogun State, Nigeria, 2 Department of Veterinary Pharmacology and Toxicology, College of Veterinary Medicine, Federal University of Agriculture, Abeokuta, Ogun State, Nigeria, 3 Department of Veterinary Public Health and Preventive Medicine, University of Ibadan, Ibadan, Oyo State, Nigeria, 4 Department of Veterinary Surgery and Theriogenology, College of Veterinary Medicine, Federal University of Agriculture, Abeokuta, Ogun State, Nigeria, 5 Department of Veterinary Microbiology, College of Veterinary Medicine, Federal University of Agriculture, Abeokuta, Ogun State, Nigeria, 6 Veterinary Council of Nigeria, Federal Capital Territory, Abuja, Nigeria, 7 Nigerian Field Epidemiology and Laboratory Training Program, Lagos, Nigeria, 8 Department of Veterinary Anatomy, College of Veterinary Medicine, Federal University of Agriculture, Abeokuta, Ogun State, Nigeria

* adebowaleoluwawemimo1@gmail.com

**Data Availability Statement:** The datasets have been deposited in Mendeley data. Adebowale, Oluwawemimo; Adenubi, Olubukola; Adesokan,

## Abstract

The novel Coronavirus SARS-CoV-2 (COVID-19) is a global pandemic with an increasing public health concern. Due to the non-availability of a vaccine against the disease, non-pharmaceutical interventions constitute major preventive and control measures. However, inadequate knowledge about the disease and poor perception might limit compliance. This study examined COVID-19-related knowledge, practices, perceptions and associated factors amongst undergraduate veterinary medical students in Nigeria. A cross-sectional web survey was employed to collect data from 437 consenting respondents using pre-tested self-administered questionnaire (August 2020). Demographic factors associated with the knowledge and adoption of recommended preventive practices towards COVID-19 were explored using multivariate logistic regression at $P \leq 0.05$. The respondents' mean knowledge and practice scores were 22.7 (SD ± 3.0) and 24.1 (SD ± 2.9), respectively with overall 63.4% and 88.8% displaying good knowledge and satisfactory practice levels. However, relatively lower proportions showed adherence to avoid touching face or nose (19.5%), face mask-wearing (58.1%), and social distancing (57.4%). Being in the 6th year of study (OR = 3.18, 95%CI: 1.62–6.26, $P$ = 0.001) and female (OR = 2.22, 95% CI = 1.11–4.41, $P$ = 0.024) were significant positive predictors of good knowledge and satisfactory practices, respectively. While only 30% of the respondents perceived the pandemic as a scam or a disease of the elites (24.0%), the respondents were worried about their academics being affected negatively (55.6%). Veterinary Medical Students in Nigeria had good knowledge and satisfactory preventive practices towards COVID-19; albeit with essential gaps in the key non-pharmaceutical preventive

Hezekiah; Oloye, Abimbola; Bankole, Noah; Fadipe, Oladotun; Ayo-Ajayi, Oluwatoyin; Akinloye, Adebayo (2020), "SARS-CoV-2 (COVID-19 Pandemic) in Nigeria: Multi-institutional Survey of Knowledge, Practices and Perception Amongst Undergraduate Veterinary Medical Students", Mendeley Data, V1, doi: 10.17632/jy7hh77f8c.1.

**Funding:** The author(s) received no specific funding for this work

**Competing interests:** The authors have declared that no competing interests exist.

measures recommended by the WHO. Therefore, there is a need to step up enlightenment and targeted campaigns about COVID-19 pandemic.

## Introduction

The COVID-19 pandemic is an ongoing infection that has spread to over 188 countries globally with over 245, 984 new cases, 25,602,665 confirmed, and 852,758 deaths as at September 2nd 2020 [1]. The disease was first reported to have originated from Wuhan, China and the causative agent identified as a novel coronavirus, Severe Acute Respiratory Syndrome (Coronavirus-2 SARS-CoV-2) [2]. This disease is similar to the previously emerged SARS-CoV and the Middle East Respiratory Syndrome Coronavirus (MERS-CoV). COVID-19 was announced as a pandemic by the World Health Organization and disease of a public health emergency globally on March 12, 2020 [2]. Subsequently, countries globally have had to implement global standard control strategies, which had hitherto not been employed since the Spanish Flu epidemic. These measures, which included travel restrictions, lockdowns or curfews, workplace hazard controls, closure of public facilities including pubs, restaurants, gyms, schools and higher institutions, strict hand hygiene practices, social distancing and the wearing of facemasks have impacted lives on a global scale. Despite these mitigation measures, the number of cases is still on the increase globally with the Americas, Europe and South-East Asia badly affected [1].

Nigeria reported its index case of COVID-19 on February 27, 2020; incidentally, the first in Nigeria and West Africa according to the Nigerian Centre for Disease Control [3]. Subsequently, a lockdown or curfew in various states was implemented to contain the fast spread of the virus. All citizens except those on essential duties were expected to stay at home and maintain good handwashing hygiene practices, local and international travels were restricted, businesses, offices, public gatherings (including religious places), schools and universities were closed, and public and private sports cancelled. According to the NCDC, more than 286,000 tests, 43,537 confirmed positive cases, 22,567 active cases, 20,087 discharges and 883 human deaths were reported as at the commencement of this study, August 1st, 2020 [3] across 36 states in the country, including the Federal Capital Territory (FCT), Abuja. However, the numbers of cases and deaths are on the rise with 56,177 confirmed cases, 1,078 fatalities as at the time of article submission (September 12th, 2020).

The pandemic has brought about huge negative consequences on business, education, health, and tourism globally [4]. Presently, primary, secondary and tertiary institutions in Nigeria are still closed and this has seriously affected millions of students in tertiary institutions who have their semesters cancelled or suspended due to the pandemic. While many other countries have switched to virtual learning, many tertiary institutions within Nigeria lack the various online educational platforms or facilities for such method of teaching [5], which could worsen the situation for students in the country. Several studies have reported students' mental health becomes greatly affected when faced with a public health emergency and academic delays, which has been positively correlated with anxiety levels [4, 6–9]. Studies by previous authors had shown that COVID-19 has a profound impact on the public, medical students, dental medical students, and radiology trainees, as well as the knowledge, practices and attitudes [10–13] but none, is known yet about veterinary medical students in Nigeria. Adequate knowledge among individuals measures the first line of defence against this disease

[14]. It is therefore important to understand the knowledge, views, adherence to the Nigerian government control policies among the veterinary student population.

To the best of our knowledge, this is the first study that would investigate the knowledge, preventive practices and perceived impacts (KPP) of COVID-19 pandemic among veterinary medical students in Nigeria.

## Materials and methods

### Study design and setting

This cross-sectional, multi-institutional web survey was conducted from August 1st to 18th, 2020 among undergraduate veterinary students in Nigeria, a West African country that is comprised of 36 states categorized into six geopolitical zones–South West, South East, South South, North East, North West, North Central and the Federal Capital Territory (FCT). The country runs the Doctor of Veterinary Medicine (DVM) programme, a six-year course in twelve universities and regulated by the Veterinary Council of Nigeria (VCN). The programme is divided into three phases namely, the preclinical (year two and three i.e. DVM 1 and 2), paraclinical (year four i.e. DVM 3) and clinical (year five and six i.e. DVM 4 and 5).

### Study population structure, sample size and sampling

The study population included veterinary medical students in 11 veterinary schools in the country. These universities and their respective geopolitical zones are University of Nigeria, Nsukka, Enugu State, Michael Okpara University of Agriculture, Umudike, Abia State (South East); Federal University of Agriculture, Makurdi, Benue State, University of Jos, Plateau State, University of Ilorin, Kwara State (North Central); Ahmadu Bello University, Zaria, Kaduna State, Usmanu Danfodiyo University, Sokoto, Sokoto State (North West); Federal University of Agriculture, Abeokuta, Ogun State, University of Ibadan, Ibadan, Oyo State ((South West); University of Maiduguri, Borno, Borno State (North East); and the University of Abuja (FCT) Fig 1.

The inclusion criteria for the participants were 1) students must be fully registered in any one of the veterinary schools previously listed, 2) and must be in DVM 1 (year 2) to DVM 5 (year 6). One university was excluded from the study as it was yet to reach the clinical phase of the veterinary programme. Similarly, Year 1 students were excluded largely because of non-exposure to core veterinary courses. A total number of all the veterinary medical students who

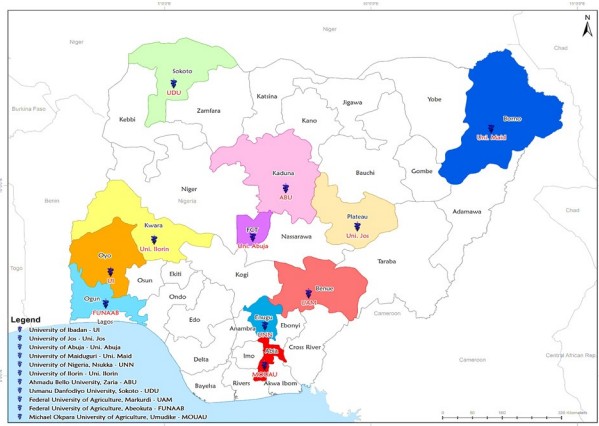

**Fig 1. The spatial distribution of veterinary schools of respondents.**

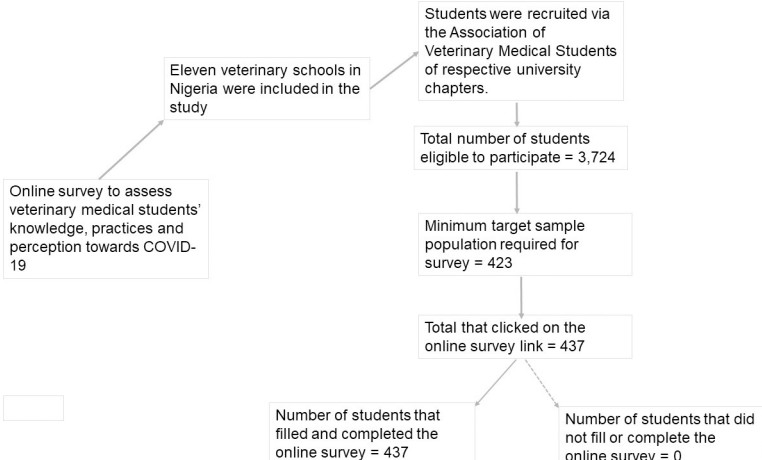

**Fig 2. The flowchart recruitment process of veterinary medical students for the online survey recruitment for the online survey.**

were eligible to participate in the online survey was 3,724. Fig 2 provides the flowchart process for student recruitment for the online survey.

The sample size for this survey was calculated based on the assumptions that poor levels of knowledge and practices among respondents were 50%, an absolute precision of 95% confidence interval, and an acceptable error of 5%. Using Working in Epidemiology (WinEpi v.2.0), a total of 385 participants was estimated while a 10% non-contingency was added to make up for non-response, giving a minimum target sample size of 423 participants [15]. The total sample size was divided equally among the veterinary schools (43 students per university). We then conveniently recruited a minimum of nine at each level (DVM 1-DVM 5) across the participating universities while participation was made voluntary. We employed an online survey due to the COVID pandemic and lockdown policy in the country as at the time of study. Briefly, prior commencement of the study, the national president of the Association of Veterinary Medical Students (AVMS) and his counterparts at the various university chapters were contacted. A detailed information on the project focus, aims, and plans for student recruitment were discussed over several online meetings. Following their consent, invitations were sent to students nationwide to participate in the study using the WhatsApp platforms of the Association of Veterinary Medical Students (AVMS) of the various university chapters. The link to the online survey questionnaire was included in the sent invitations and a brief description of the purpose of the study was provided. Also, class coordinators at various levels (DVM 1 to 5) were further assigned to share the questionnaire on their respective class WhatsApp platforms, while three of the authors were delegated to follow up on this process to enhance participation. Also, call credit top up cards were offered as incentives for participation and completion of the survey.

## Questionnaire design and pretest

The questionnaire using google forms (Alphabet Inc., California, USA) comprised a total of 41 questions (both open and closed-ended) written in English and adapted from WHO resources and other pertinent studies [2, 4, 16–20]. The questionnaire was divided into four sections and comprised questions on students' demographics, knowledge, practices, and perceptions towards COVD-19.

The first section consisted of questions assessing the socio-demographic profiles of the respondents (further considered as our independent variables). These included age as at last birthday (in years), sex, religion, the name of Institution, programme year, state of residence during the lockdown, number of household members, and type of lockdown instituted by the state government where resident.

To measure students' general knowledge about COVID-19, an 11-item questionnaire that assessed the source of information about COVID-19 and general awareness questions were provided in section B. Question one addressed the various sources of updates and information on the pandemic were requested. The other ten questions focused on clinical presentation, transmission, prevention, and control strategies of COVID-19. Question asked included the cause of COVID-19 infection, incubation period, risk conditions, country of the first outbreak, modes of transmission, identification of common symptoms of the infection, if it was possible to have asymptomatic individuals, and methods of prevention, control and treatment. Each correct answer weights 1 point and 0 for incorrect or I don't know answers. Score for each responses was summed up to give 31 points.

Section C of the questionnaire assessed respondents' practices during the pandemic and comprised 3 Likert-item questions (10) were adopted from recommended guidelines of the WHO and Ministry of Health, Nigeria for the prevention of COVID-19 transmission. These included handwashing/sanitizing, avoiding crowded places, keeping physical/social distance, avoiding touching of face or nose, avoiding handshakes, use of facemasks, and medications. The responses were never, sometimes, all the time each weighing 1, 2, and 3 points respectively. The Score for each response was summed up to give 30 points.

The questions in section D were structured to evaluate respondents' perceptions. The 5-point Likert item questions (12) were designed to assess students' perceptions of the infection based on the country's peculiarity. Some of the questions asked included whether the disease was a scam, affected only the elites, impact on academics and virtual learning in higher institutions, stigmatization, whether participants were optimistic the pandemic would be brought under control, and if they felt depressed. The agreement scale ranged from '1' for "strongly agree" to '5' for "strongly disagree".

The questionnaire was reviewed by a panel of experts and revised based on their comments. Subsequently, it was pilot-tested (n = 13 students from all the eleven veterinary schools, who were excluded in the main study), to check for its applicability and clarity before commencement of the study. All the necessary modifications were done based on outcome of the pilot study. The completion of the online survey took about 8 minutes and designed to ensure duplicate entries was avoided by preventing users with the same IP address access to the survey twice in the google form settings. Detailed information on the questionnaire is presented in S1 File.

The online survey was conducted based on the Checklist for Reporting Results of Internet E-Surveys (CHERRIES), and guidelines for good practice in the conduct and reporting of online research [21].

## Ethical approval

The study protocol was approved by the College of Veterinary Medicine, Federal University of Agriculture Research Ethics Committee (reference number: FUNAAB/COLVET/CREC/ 2020/07/01). After providing detailed description of the study and before invitations, informed consent was obtained from the presidents of the different AVMS chapters. Participants' consent was obtained verbally and witnessed by the class coordinators of the various levels (HOCs–DVM 1 to 5). Participation in the study was voluntary without any attached

penalty for refusal; personal identifiers were not collected and information from respondents was treated confidentially. Every participant was notified of his/her right to discontinue at any stage of the study according to the World Medical Association Declaration of Helsinki, 2001 [22].

## Data analysis

Data generated were captured and filtered in Microsoft Excel®, 2013 (Microsoft Corporation, Redmond, WA). Data analyses were conducted by GraphPad Prism 8.0.0 (descriptive statistics and figure presentations) and Stata 12.0 (inferential statistics). Descriptive statistics were conducted for all variables and presented in forms of frequencies and proportions/percentages using Microsoft Excel® (2013). As for the descriptive statistical methods, the following were used: measures of central tendency (arithmetic mean and median), measures of variability (standard deviation), and as absolute numbers (n) and percentage representation. To evaluate the knowledge level of respondents, a numeric pattern of scoring was used by giving a score of "1" for the "correct answer" and "0" for an "incorrect" or "I don't know" response. Similarly, the practice level was assessed by giving scores of "1" for the "never" and "2" for "sometimes" and 3 for "all the time" responses. The levels of measured outcomes were expressed as mean and standard deviation (Mean ± SD). The measured outcomes were tested for normality using the using Kolmogorov-Smirnov (> 0.05), which informed our use of (Mean ± SD). The scores were thereafter converted to percentages, and based on the students' mean scores in knowledge and practices categories, cut-off points for good / satisfactory were set at ≥70%, while those below (<70%) were considered to have poor/ unsatisfactory levels. These cut-off points were so set since it is expected that such students on medical profession should have basic knowledge and demonstrate practices towards issues related to health. Besides, such cut-off points had earlier been employed in a similar study [23]. Mean scores were compared across demographic categories using ANOVA and independent t-tests where appropriate. For post hoc comparision Dunnett's test was performed.

Associations between the socio-demographics of respondents (independent variables) and binary outcomes of knowledge and practices (dependent variables) using chi-square tests were determined. To determine potential predictors influencing knowledge and practice levels towards COVID-19 prevention among undergraduate veterinary students in Nigerian universities, outcomes significant at p ≤ 0.25 at the univariate analysis were processed further by a stepwise forward likelihood multivariate analysis (logistic regression model) using Stata 12.0 was performed. This was chosen in order to avoid variation in results from individual univariate tests of different measures due to random chance. The decision for a liberal p-value (p ≤ 0.25) at this step was to ensure important potential predictor/risk variables were included in the model. A p<0.05 was considered statistically significant and odds ratios were computed to determine the strength of associations between variables at 95% confidence intervals (CIs). All illustrations were performed with GraphPad Prism 8.0 and Microsoft Excel®.

## Results

### Demographic information of respondents

A total of 437 respondents participated in this study across Nigeria, which provided a participation rate of more than 100.0%. The mean age of respondents was 22.14 ± 2.99 years, and the median number of members in a household was 6 (min = 1, max = 90). The highest percentage of the respondents (17.2%, 95% CI; 13.9–21.0) were in the Federal University of Agriculture, Abeokuta and DVM 4 (year 5, 23.8%, 95% CI; 20.0–28.0). Majority of respondents resided in

the South West (38.9%, 95% CI; 34.6–43.7) and North Central (38.2%, 95% CI; 33.7–42.8) regions of the country. The demographics of the study sample are presented in Table 1.

**Table 1. Socio-demographic profile of veterinary medical students in Nigeria that participated in the online survey.**

| | Variable | Category | Proportions |
|---|---|---|---|
| 1. | Age (in years) | 16–20 | 138 (31.7) |
| | | 21–25 | 249 (57.1) |
| | | 25–30 | 43 (9.9) |
| | | 31–35 | 5 (1.1) |
| | | >35 | 1 (0.2) |
| 2. | Sex | Male | 260 (59.5) |
| | | Female | 177 (40.5) |
| 3. | Religion | Christianity | 316 (72.3) |
| | | Islam | 119 (27.2) |
| | | Others | 2 (0.5) |
| 4. | Name of University | FUNAAB | 75 (17.2) |
| | | UI | 73 (16.7) |
| | | UNILORIN | 26 (5.9) |
| | | UNIABUJA | 71 (16.2) |
| | | UNIJOS | 27 (6.2) |
| | | UNIMAID | 18 (4.1) |
| | | UDUS | 47 (10.8) |
| | | ABU | 14 (3.2) |
| | | MOUAU | 8 (1.8) |
| | | UAM | 60 (13.7) |
| | | UNN | 18 (4.1) |
| 5. | Level/Year of study | DMV 1 (year 2) | 92 (21.1) |
| | | DVM 2 (Year 3) | 83 (19.0) |
| | | DVM 3 (Year 4) | 82 (18.8) |
| | | DVM 4 (Year 5) | 104 (23.8) |
| | | DVM 5 (Year 6) | 76 (17.8) |
| 6. | No. of household members | Less than 5 | 96 (22.3) |
| | | 5–10 | 295 (68.4) |
| | | >10 | 40 (9.3) |
| 7. | Geopolitical region | North Central | 166 (38.2) |
| | | North East | 25 (5.7) |
| | | North West | 40 (9.2) |
| | | South East | 21 (4.8) |
| | | South South | 13 (3.0) |
| | | South West | 170 (39.1) |
| 8. | Type of lockdown | Partial | 299 (68.4) |
| | | Total | 103 (23.6) |
| | | Not Sure | 35 (8.0) |

FUNAAB—Federal University of Agriculture, Abeokuta; UI—University of Ibadan, Ibadan; UNILORIN—University of Ilorin, Ilorin; UNIABUJA—University of Abuja, Abuja; UNIJOS—University of Jos, Jos; UNIMAID—University of Maiduguri, Maiduguri; UDUS—Usmanu Danfodiyo University, Sokoto; ABU—Ahmadu Bello University, Zaria; MOUAU—Michael Okpara University of Agriculture, Umudike; UAM—University of Agriculture, Makurdi; UNN —University of Nigeria, Nsukka.

## Respondents knowledge level towards COVID– 19

The most preferred source was through social media/internet platforms (n = 403), including Facebook, Instagram, and Twitter. The less employed sources of information were newspapers (n = 126) and classrooms (n = 98).

The respondents showed an overall mean knowledge score of 22.7 (SD ± 3.0; score 0 → 31), suggesting a mean level of 73.4% (SD ± 9.7%, range 38.7–93.5%) on COVID– 19. A total of 277 (63.4%) students had knowledge scores ≥ 70% cut off, however, none responded correctly to all the knowledge items (71–93.5%). Knowledge varied significantly among the age groups (p = 0.0042) with those within groups 21–25 years and 26–30 years having higher scores (p = 0.011). Similarly, knowledge scores significantly varied among DVM levels (p = 0.0001) and Veterinary schools (p = 0.027) with students in DVM 5 (Year 6) outperforming those in the preclinical levels DVM 1 (p<0.0001), DVM 2 (p = 0.024) and DVM 3 (p = 0.042).

High proportions of students correctly identified COVID– 19 as a viral infection (98.9%, n = 432) and that it originated from Wuhan China (99.3%, n = 434), while 27.9% correctly (n = 122) reported it as similar to both SARS-CoV and MERS-CoV. Clinical signs associated with COVID-19 as identified by respondents were as follows: fever (97.5%, n = 426), fatigue (70.9%, n = 310), dry cough (86.5%, n = 378), runny nose (38.0%, n = 166), shortness of breath (91.5%, n = 400), myalgia (35.0%, n = 153), loss of taste (40.0%, n = 175), loss of smell (46.0%, n = 201), and diarrhoea/vomiting (21.3%, n = 93). Only 5 (1.1%) of the respondents correctly identified all the possible clinical presentations. A high proportion (84.2%, n = 368) knew that the COVID-19 virus spreads via respiratory droplets of infected people, and asymptomatic state of infection and transmission is possible (88.3%, n = 386). Also, majority of the respondents knew the application of alcohol-based sanitizers (95.9%), soap and detergent (75.0%), high temperature inactivates or kills the virus (61.1%). Meanwhile, 308 (70.5%) were aware there was no cure for the disease. Table 2 presents details of knowledge components and students' performance on various questions on COVID-19.

## COVID-19 and self-reported preventive practices of respondents

Majority of the students reported maintaining good personal hygiene (n = 386, 84.2%), while a lower proportion would not touch their face or nose all the time (n = 85, 19.5%). Averagely, respondents observed the stay at home policy (n = 219, 50.1%), face mask-wearing in public (n = 254, 58.1%), and social distancing from people (n = 251, 57.4%). Up to 66.4% (n = 299) reported never self-medicating to prevent COVID- 19 infection. A significant association between the knowledge that there was no cure for COVID-19 and not self- medicating was observed (p = 0.01). Fig 3 further describes in details respondents' practices towards preventing being infected and community spread.

The overall preventive practice mean score of students towards COVID- 19 was 24.1 (SD ± 2.9; score 0 → 30), suggesting a mean level of 80.3% (SD ± 9.6%, range 40.0–100.0%). The practice level was generally satisfactory with 88.8% (n = 388) of the respondents having the ≥ 70% cut off, while one student reported observing all the preventive measures. A positive correlation between preventive practice measures and knowledge about COVID– 19 was observed, although weak (r = 0.16, n = 437, p = 0.0009, 95% CI; 6.2–25.0). The practice scores were similar across DVM levels (p = 0.09), geopolitical regions of residence (p = 0.36), and lockdown type (p = 0.10).

**Table 2. Knowledge of COVID-19 among veterinary medical students in Nigeria, August 2020 (n = 437).**

| Knowledge items and correct answers | Frequency (%) | |
|---|---|---|
| | **Correct** | **Incorrect** |
| **1.** COVID-19 is caused by a virus | 432 (98.9) | 5 (1.10 |
| **2.** Incubation period of the disease is 2–14 days | 410 (93.8) | 27 (6.2) |
| **3.** COVID- 19 is similar to MERS-CoV | 134 (30.7) | 303 (69.3) |
| SARS-CoV | 288 (65.9) | 149 (34.1) |
| **4.** COVID- 19 was first reported in China | 434 (99.3) | 3 (0.7) |
| **5.** Possible common symptoms of COVID- 19: Fever | 426 (97.5) | 11 (2.5) |
| Dry cough | 378 (86.5) | 59 (13.5) |
| Runny nose | 166 (38.0) | 271 (62.0) |
| Shortness of breath | 400 (91.5) | 37 (8.5) |
| Joint/Muscle ache | 153 (35.0) | 284 (65.0) |
| Loss of taste | 175 (40.0) | 262 (60.0) |
| Loss of smell | 201 (46.0) | 236 (54.0) |
| Diarrhoea/vomiting | 93 (21.3) | 344 (78.7) |
| Fatigue | 310 (70.9) | 127 (29.1) |
| **6.** COVID 19 can be transmitted through Direct contact with an infected person | 401 (91.8) | 36 (8.2) |
| Air droplet | 368 (84.2) | 69 (15.8) |
| Indirect contact such as contaminated surfaces | 285 (65.2) | 152 (34.8) |
| Handshake | 357 (81.7) | 80 (18.3) |
| Kissing | 303 (69.3) | 134 (30.7) |
| **7.** It is possible to have COVID 19 and not show symptoms | 386 (88.3) | 51 (11.7) |
| **8.** What can kill the virus: Alcohol-based sanitizers | 419 (95.9) | 18 (4.1) |
| Soap/detergents | 328 (75.1) | 109 (24.9) |
| High-temperature application | 267 (61.1) | 170 (38.9) |
| **9.** Pets have been scientifically proven to transmit COVID- 19 | 184 (42.1) | 253 (57.9) |
| **10.** There is a cure for COVID 19 | 308 (70.5) | 129 (29.5) |

## Bivariate analysis for the association between sociodemographic profiles of respondents and their knowledge and practice levels on COVID-19 pandemic

The bivariate analysis showed that only age (p = 0.017) and year of study (p = 0.009) were significantly associated with knowledge levels at p $\leq$ 0.25 (Table 3).

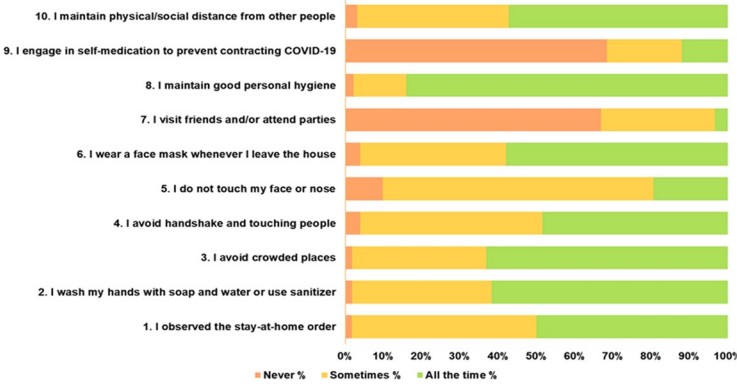

**Fig 3. Self–reported practices by veterinary medical students in Nigeria towards preventing infection and community spread during the COVID– 19 pandemic.**

**Table 3. Bivariate analysis of veterinary medical students' socio-demographics with the knowledge levels about COVID-19 pandemic in Nigeria.**

| Variable | Category | Knowledge level | | $X^2$ | P value |
|---|---|---|---|---|---|
| | | Good | Poor | | |
| Age (in years) | 16–20 | 79 (57.2) | 59 (42.8) | | |
| | 21–25 | 158 (63.4) | 91 (36.6) | 8.12 | 0.017* |
| | ≥25 | 40 (80.0) | 10 (20.0) | | |
| Sex | Male | 164 (63.1) | 96 (36.9) | | |
| | Female | 113 (63.8) | 64 (36.2) | 0.03 | 0.871 |
| Religion | Christianity | 202 (63.7) | 115 (36.3) | | |
| | Islam | 75 (62.5) | 45 (37.5) | 0.06 | 0.813 |
| Level/Year of study | DMV 1 (year 2) | 48 (52.2) | 44 (47.8) | | |
| | DVM 2 (Year 3) | 48 (57.8) | 35 (42.2) | | |
| | DVM 3 (Year 4) | 52 (63.4) | 30 (36.6) | 13.42 | 0.009* |
| | DVM 4 (Year 5) | 70 (67.3) | 34 (32.7) | | |
| | DVM 5 (Year 6) | 59 (77.6) | 17 (22.4) | | |
| No. of household members | Less than 5 | 120 (62.2) | 73 (37.8) | | |
| | 5–10 | 129 (63.9) | 73 (36.1) | | |
| | >10 | 28 (66.7) | 14 (33.3) | 0.34 | 0.845 |
| Geopolitical region | North Central | 109 (65.7) | 57 (34.3) | | |
| | South West | 108 (63.5) | 62 (36.5) | | |
| | **Others | 60 (59.4) | 41 (40.6) | 1.06 | 0.588 |
| Type of lockdown | Partial | 196 (65.6) | 103 (34.4) | | |
| | Total | 60 (58.2) | 43 (41.8) | 1.95 | 0.378 |
| | Not Sure | 21 (60.0) | 14 (40.0) | | |
| Total | | 277 (63.4) | 160 (36.6) | | |

* $P \leq 0.25$;

**Others: North east, North west, South east, South south

Similarly, sex (p = 0.012), religion (p = 0.076), and geopolitical region (p = 0.022) were associated factors with practice level of respondents towards COVID-19 pandemic (Table 4).

**Multivariate analysis for the association between sociodemographic profiles of respondents and their knowledge and practice levels on COVID-19 pandemic.** The multivariate logistic regression analysis reveals only the level/year of study (p = 0.014) and sex (p = 0.024) of respondents respectively were significant positive predictors of good knowledge and practice levels towards COVID-19. Respondents in Year 6 (clinical) were about 3.2 times more likely to have good knowledge of COVID-19 pandemic (OR = 3.18, 95%CI: 1.62–6.26, p = 0.001) than those in Year 2 (non-clinical). On the other hand, the female had higher odds of demonstrating satisfactory practices regarding COVID-19 pandemic (OR = 2.22, 95%CI = 1.11–4.41, p = 0.024) than the males (Table 5). Besides, respondents in the regions marked others (OR = 0.37, 95%CI: 0.17–0.78, p = 0.009) had significantly lowest odds of demonstrating satisfactory practices regarding COVID-19 (Table 5).

## Perceptions about COVID– 19

Most of the respondents reported they have seen persons infected with COVID-19 and do not think the pandemic was a scam (68.6%, n = 300), or a disease of the elites (76.0%, n = 332). Also, 73.9% (n = 323) respondents disagreed they had internet facilities to

**Table 4. Bivariate analysis of veterinary medical students' sociodemographic with the practice levels about COVID-19 pandemic in Nigeria.**

| Variable | Category | Practice level | | X$^2$ | P value |
|---|---|---|---|---|---|
| | | Satisfactory | Unsatisfactory | | |
| Age (in years) | 16–20 | 119 (86.2) | 19 (13.8) | | |
| | 21–25 | 224 (90.0) | 25 (10.00) | 0.61 | 0.737 |
| | ≥25 | 44 (88.0) | 6 (12.0) | | |
| Sex | Male | 222 (85.4) | 38 (14.6) | | |
| | Female | 165 (93.2) | 12 (6.8) | 6.38 | 0.012* |
| Religion | Christianity | 286 (90.2) | 31 (9.8) | | |
| | Islam | 101 (84.2) | 19 (15.8) | 3.15 | 0.076* |
| Level/Year of study | DMV 1 (Year 2) | 79 (85.9) | 13 (14.1) | | |
| | DVM 2 (Year 3) | 73 (88.0) | 10 (12.0) | | |
| | DVM 3 (Year 4) | 74 (90.2) | 8 (9.8) | 1.06 | 0.901 |
| | DVM 4 (Year 5) | 93 (89.5) | 11 (10.5) | | |
| | DVM 5 (Year 6) | 68 (89.5) | 8 (10.5) | | |
| No. of household members | Less than 5 | 174 (90.2) | 19 (9.8) | | |
| | 5–10 | 177 (87.6) | 25 (12.4) | 1.00 | 0.608 |
| | >10 | 36 (85.7) | 6 (14.3) | | |
| Geopolitical region | North Central | 154 (92.8) | 12 (7.2) | 7.67 | 0.022* |
| | South West | 150 (88.2) | 20 (11.8) | | |
| | **Others | 82 (81.2) | 19 (18.8) | | |
| Type of lockdown | Partial | 264 (88.3) | 35 (11.7) | | |
| | Total | 93 (90.3) | 10 (9.7) | 0.61 | 0.739 |
| | Not Sure | 30 (85.7) | 5 (14.3) | | |
| Total | | 387 (88.8) | 50 (11.2) | | |

* Significant at $P \leq 0.25$;

**Others: North east, North west, South east, South south

educate themselves with online programmes related to the profession. Averagely, 55.6% (n = 243) students were worried their academic performance would be affected negatively and 50.1% (n = 199) spend more time on social media than studying. A good proportion (71.6%, n = 313) of the students were optimistic about the pandemic being over soon while 44.1% agreed being depressed as a result of the pandemic (Fig 4).

## Discussion

This study, which aptly reflected the KPP of Veterinary medical student towards the COVID-19 pandemic, is the first in Nigeria and Africa to the best of the authors' knowledge. The majority of the students had good knowledge and satisfactory practices regarding COVID-19 pandemic; however, there were important gaps in the key non-pharmaceutical preventive measures recommended by the WHO with implications for public health and disease control.

The study revealed a higher number of male respondents than females. This is similar to previous studies which showed a reflection of male dominance in the veterinary profession in Nigeria [15–17, 24]. Female veterinary students who participated in this study (40.5%), when compared with a similar and recent study conducted among veterinary professionals with 27.2% female respondents seems higher [17]. Presently, gender shift with more females than males in the veterinary profession especially in the Western world [25–28] and in South Africa [29] is reported.

**Table 5. Multivariate logistic regression analysis of factors associated with knowledge and practice levels on COVID-19 pandemic among veterinary medical students in Nigeria.**

| Variable | Category | Knowledge level | | | Practice | | |
|---|---|---|---|---|---|---|---|
| | | AOR | 95%CI | P value | AOR | 95%CI | P value |
| Age (in years) | 16–20 | 1 | | | | | |
| | 21–25 | 1.30 | 0.85–1.98 | 0.230 | | | |
| | ≥25 | 2.99 | 1.38–6.46 | 0.005 | | | |
| Sex | Male | | | | 1 | | |
| | Female | | | | 2.22 | 1.11–4.41 | 0.024* |
| Religion | Christianity | | | | 1 | | |
| | Islam | | | | 0.63 | 0.34–1.20 | 0.159 |
| Level/Year of study | DMV 1 (year 2) | 1.00 | | | | | |
| | DVM 2 (Year 3) | 1.26 | 0.69–2.29 | 0.453 | | | |
| | DVM 3 (Year 4) | 1.59 | 0.87–2.92 | 0.135 | | | |
| | DVM 4 (Year 5) | 1.89 | 1.06–3.37 | 0.032 | | | |
| | DVM 5 (Year 6) | 3.18 | 1.62–6.26 | 0.001 | | | |
| Geopolitical region | North Central | | | | 1 | | |
| | South West | | | | 0.72 | 0.34–1.52 | 0.384 |
| | **Others | | | | 0.37 | 0.17–0.78 | 0.009* |

* Significant at $P \leq 0.05$, AOR = Adjusted Odds Ratio, 1 = Reference

The respondents' mean knowledge score was 22.7 (SD ± 3.0; 73.4%) with an overall 63.4% displaying good knowledge which seems satisfactory and similar to reports for medical students in Jordan with overall 69.5% showing good knowledge [30] and veterinary professionals in Nigeria (64.0%, [17]). The knowledge level displayed was higher than reports from two university communities in Pakistan (50.2%, [31]) and Nigeria (59.5%, [16]) as well as among the public visiting a medical centre in Ethiopia (41.3%, [19]). However, other studies have recorded higher knowledge level towards the COVID- 19 pandemic among undergraduate students in China (82.3%, [32]), residents in North central Nigeria (99.5%, [33]), the United States (80%, [34]), and China (90.0%, [35]).

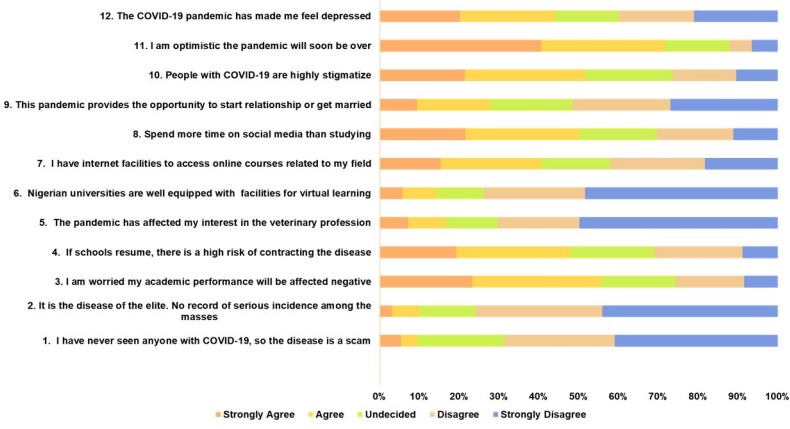

**Fig 4. The perceptions of veterinary medical students in Nigeria towards COVID-19 pandemic and the impact on their learning and emotional well being.**

Majority of the veterinary medical students relied on the internet and social media to get information or updates about the pandemic, which might have contributed to the high level of knowledge acquired about COVID-19. This follows similar studies were the internet/social media was reported as the most common source of information for medical and non-medical students [10, 18, 30]. Several guidelines and information on COVID-19 have been uploaded online by WHO and NCDC immediately after pronouncing the guidelines. Also, the public receives constant notifications or reminders about these guidelines by network service providers in the country. It is, however important that government agencies should work towards dispelling misinformation, misconceptions, rumours or hoax news from illicit social media platforms, which have increased 50 times more during the pandemic [4].

Further, being students in their final year (which corresponds to the clinical year) than in non-clinical was a positive predictor for good knowledge about COVID- 19. This is not surprising because all veterinary schools engage students in clinical courses in veterinary public health, epidemiology of infectious/zoonotic diseases, mechanisms of disease spread and control. It is expected that the greater the exposure to clinical teaching in the final year of veterinary school, the more the students at this level will likely be keen or inquisitive to acquire more information or knowledge than others. Besides, respondents in the North West region had the lowest odds of having good knowledge of COVID-19 pandemic when compared with other geopolitical regions. This finding is very vital to planning an informed disease mitigation programme as such data are required in enhancing targeted educational programmes among university student populations in the country. Multi-stakeholder collaborative efforts and strategies should be promoted within institutions to contain pandemic among university students.

The level of preventive practices among the students (88.8%) was also commendable; however, there were some important gaps in public health concerns. Averagely, respondents (50.1%) observed the stay at home policy, face mask-wearing in public (58.1%), and social distancing from people (57.4%). While these three constitute key non-pharmaceutical preventive measures recommended by the WHO, it appears worrisome that only a little above average of the respondents adhered to these measures. Meanwhile, the issue of face masking is becoming debatable within some groups of people globally. Some see the policy as a bridge of human rights, while some other people feel uncomfortably hot and experience difficulty in breathing when wearing face masks. Our data on face masking is similar to other studies which showed that although people know it is one of the protective guidelines, many do not frequently comply with its use in public places [18, 31, 33, 34, 36]. Now that Colleges and Universities will soon be re-opened in Nigeria, targeted education and measures should be in place to ensure students comply with the key protective guidelines especially the wearing of face masks on resumption.

Majority of the respondents practiced good hygiene and did not use any self-medication as prophylaxis. It was observed that knowing that there was no cure for the virus significantly influenced the respondents' choice not to self-medicate as a prophylactic measure. Female students displayed a satisfactory practice level twice more than male in this study. This is not surprising as the females are viewed to be more cautious than their male counterparts who often dare and take risks. Findings from Pakistan showed literate society, particularly women had good knowledge, optimistic attitudes, and practices towards COVID-19 [37]. Similar studies conducted in China and Pakistan showed preventative practices were better in the female population than males [31, 35]. Other studies, however, reported good practices were associated with age, gender and education [37, 38].

Again, the majority of the respondents had right perceptions about COVID-19 pandemic, as 76.0% perceived the disease as not a scam nor a disease of the elites (70.0%). Besides, the majority of the students held an optimistic attitude with 71.6% believing that COVID-19 would finally be successfully controlled. Such perceptions could promote global drive at containing the pandemic

since this might eventually rub on their level of adherence to preventive measures. Importantly, the consequent perceived impacts of COVID-19 pandemic on academics in Nigeria by the respondents is a matter of concern. The respondents were worried their academic performance would be affected negatively (55.6%) and that they spent more time on social media than studying (50.1%). Many (74.0%) shared their dissatisfaction about the government tertiary institutions because of the inadequate online facilities or tools to perform virtual education in Nigeria during such a time like this. While many colleges and universities worldwide switch to online teaching to reduce people contact, public universities in Nigeria have not been able to achieve this.

Lastly, over 44.0% of student participants indicated having depression due to the pandemic. Several studies have shown the psychological impact of the epidemic on the general public, patients, medical staff, children, and older adults [6, 39, 40]. Students' mental health is greatly affected and may worsen existing mental health problems when faced with a public health emergency, social isolation, and economic recession. In times like this, students need attention, assistance, and support from the community, family, and tertiary institutions [4]. The emotional status of the respondents, as reflected by their response to their perception about marital issues appears to favour disinterest in love relationship and marriage contraction. This might be attributed to more engaging thoughts about overcoming the prevailing pandemic.

Some limitations of the approach utilized in the study were identified. The introduction of enrolment and reporting biases may have resulted from the online survey making it non-representative. For the study, randomization was impossible due to the national lockdown, which could have possibly eliminated some of the biases. The non- probabilistic sampling approach, which is convenience and voluntary may have contributed to in the uneven distribution (coverage and participation) of student respondents from the different universities investigated. Furthermore, the poor internet accessibility or connectivity in the country (which may have varied from one institution location to another) and lack of funds to purchase data as complained by some students may have contributed to the lack of access to online questionnaire and participation. We are therefore cautious in generalizing the sample findings to the whole veterinary student populations in the country due to these limitations.

## Conclusion

Although the knowledge and preventive practices of the veterinary students in this study were satisfactory, there were important gaps in some key preventive practices recommended by the WHO. Some of the identified KPP gaps in this study require urgent attention and must be targeted towards promoting strategic educational planning and behavioral changes. Also, e-learning facilities should be provided within the Nigerian universities, which must be constantly upgraded and usage maximized by staff and students where necessary to promote physical distancing as much as possible.

## Supporting information

**S1 File. Questionnaire on the knowledge, practices and perception of undergraduate veterinary students towards COVID-19 in Nigeria.**
(DOCX)

## Acknowledgments

We are grateful to the national and chapter presidents of the AVMS for their commitment towards the success of this project. The authors are grateful to all the veterinary students across Nigeria who responded to the survey.

## Author Contributions

**Conceptualization:** Oluwawemimo Oluseun Adebowale, Olubukola Tolulope Adenubi.

**Data curation:** Oluwawemimo Oluseun Adebowale, Hezekiah Kehinde Adesokan, Noah Olumide Bankole, Patience Oluwatoyin Ayo-Ajayi.

**Formal analysis:** Oluwawemimo Oluseun Adebowale, Hezekiah Kehinde Adesokan.

**Investigation:** Oluwawemimo Oluseun Adebowale, Olubukola Tolulope Adenubi, Hezekiah Kehinde Adesokan, Abimbola Adetokunbo Oloye, Noah Olumide Bankole, Oladotun Ebenezer Fadipe, Patience Oluwatoyin Ayo-Ajayi.

**Methodology:** Oluwawemimo Oluseun Adebowale, Olubukola Tolulope Adenubi, Hezekiah Kehinde Adesokan, Abimbola Adetokunbo Oloye, Noah Olumide Bankole, Oladotun Ebenezer Fadipe, Patience Oluwatoyin Ayo-Ajayi.

**Project administration:** Oluwawemimo Oluseun Adebowale, Olubukola Tolulope Adenubi, Adebayo Koyuum Akinloye.

**Supervision:** Adebayo Koyuum Akinloye.

**Validation:** Adebayo Koyuum Akinloye.

**Writing – original draft:** Oluwawemimo Oluseun Adebowale, Hezekiah Kehinde Adesokan.

**Writing – review & editing:** Oluwawemimo Oluseun Adebowale, Olubukola Tolulope Adenubi, Hezekiah Kehinde Adesokan, Abimbola Adetokunbo Oloye, Noah Olumide Bankole, Oladotun Ebenezer Fadipe, Patience Oluwatoyin Ayo-Ajayi, Adebayo Koyuum Akinloye.

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
