## [Decision Letter · Decision Letter 0]

27 Oct 2020

PONE-D-20-28765

SARS-CoV-2 (COVID-19 pandemic) in Nigeria: Multi-institutional survey of knowledge, practices and perception amongst undergraduate veterinary medical students”

PLOS ONE

Dear Dr. Adebowale,

Thank you for submitting your manuscript to PLOS ONE. After careful consideration, we feel that it has merit but does not fully meet PLOS ONE’s publication criteria as it currently stands. Therefore, we invite you to submit a revised version of the manuscript that addresses the points raised during the review process.

Although two of the reviewers were more positive, there are specific methodological issues to be addressed before acceptance. For example, the power calculation, the recruitment procedures, decisions on cutpoints and scale transformation, and the rational for the multivariate analysis are not clear. Also, the discussion of the limitations of the approach utilized for the study has also not been broadly contextualized. It will be nice if you can addressed these together with the other concerns of the reviewers in your revision.

We look forward to receiving your revised manuscript.

Kind regards,

Adewale L. Oyeyemi, Ph.D

Academic Editor

PLOS ONE

Journal Requirements:

2.Thank you for including your ethics statement: 

"    The study protocol was approved by the College of Veterinary Medicine, Federal University of Agriculture Research Ethics Committee (reference number: FUNAAB/COLVET/CREC/2020/07/01). Before invitations, informed consent was obtained from the presidents of the different AVMS chapters. Participation in the study was voluntary; personal identifiers were not collected and information from respondents was treated confidentially. Every participant was notified of his/her right to discontinue at any stage of the study according to the World Medical Association Declaration of Helsinki, 2001 ".   

Please provide additional details regarding participant consent. In the ethics statement in the Methods and online submission information, please ensure that you have specified what type you obtained (for instance, written or verbal, and if verbal, how it was documented and witnessed). If your study included minors, state whether you obtained consent from parents or guardians. If the need for consent was waived by the ethics committee, please include this information.

3.We note that you have indicated that data from this study are available upon request. PLOS only allows data to be available upon request if there are legal or ethical restrictions on sharing data publicly. For information on unacceptable data access restrictions, please see http://journals.plos.org/plosone/s/data-availability#loc-unacceptable-data-access-restrictions.

4.We note that [Figure(s) 1] in your submission contain [map/satellite] images which may be copyrighted. All PLOS content is published under the Creative Commons Attribution License (CC BY 4.0), which means that the manuscript, images, and Supporting Information files will be freely available online, and any third party is permitted to access, download, copy, distribute, and use these materials in any way, even commercially, with proper attribution. For these reasons, we cannot publish previously copyrighted maps or satellite images created using proprietary data, such as Google software (Google Maps, Street View, and Earth). For more information, see our copyright guidelines: http://journals.plos.org/plosone/s/licenses-and-copyright.

1.    You may seek permission from the original copyright holder of Figure(s) [1] to publish the content specifically under the CC BY 4.0 license. 

5. Please upload a new copy of Figure 1 as the detail is not clear. Please follow the link for more information: https://blogs.plos.org/plos/2019/06/looking-good-tips-for-creating-your-plos-figures-graphics/" https://blogs.plos.org/plos/2019/06/looking-good-tips-for-creating-your-plos-figures-graphics/

Reviewers' comments:

Reviewer's Responses to Questions

**Comments to the Author**

1. Is the manuscript technically sound, and do the data support the conclusions?

Reviewer #1: Partly

Reviewer #2: Partly

Reviewer #3: Yes

2. Has the statistical analysis been performed appropriately and rigorously? 

Reviewer #1: Yes

Reviewer #2: No

Reviewer #3: Yes

3. Have the authors made all data underlying the findings in their manuscript fully available?

Reviewer #1: Yes

Reviewer #2: Yes

Reviewer #3: Yes

4. Is the manuscript presented in an intelligible fashion and written in standard English?

Reviewer #1: Yes

Reviewer #2: Yes

Reviewer #3: Yes

5. Review Comments to the Author

Reviewer #1: This study is well conducted, ethically sound and relevant as it provides insight on the knowledge & Practices of this important Pandemic in the study population. I have a few comments and suggestions.

1) Line 78 “However, the numbers of cases and deaths are still on the rise…….” is this statement correct as at the time of writing????

2) Line 127-128 “At least, it was ensured one veterinary school was representative of each of the geopolitical zones in Nigeria” …… this statement is redundant as the only university that was not represented was the one that was excluded.

3) Lines 129-131 “A total of 385 participants were needed considering ………. with an acceptable error of 5%” ……….. were needed to achieve what??? Power????

4) Lines 131-132 “ 10% of the sample size was added to give a total of 423 participants …. 10% attrition = 478

5) Although it was stated in line 132 that ……” For student recruitment, a purposive and chain referral sampling techniques was performed” give details of how the sampling was done. What is the end point? How was the sample size distributed for the variuod schools etc

6) Line 150 “responses were scored” How???

7) Lines 190-192 “The levels of measured outcomes were expressed as mean and standard deviation (Mean ± SD)”……. Even if they are not normally distributed????

8) Line 197 “Variables significant at p ≤ 0.25 were processed further by the logistic regression” what inform the use of this cut off

9) I don’t know if the 95% CI in table 1 is necessary

10) The OR in table 5is it crude or adjusted????

11) The first sentence in the conclusion is not derived from your data

Reviewer #2: Line 130: In the methods, the required sample size was about 400. This is a standard for a KAP Questionnaire study. However, the sample size does not require response rate estimates. It requires the expected differences to be detected. Please reconsider this note in the writing up.

Line 133: “For student recruitment, a purposive and chain referral sampling 133 techniques was performed”. A brief description may be necessary as this is not a standard sampling method used for questionnaires.

Line 145:these questions are not clearly presented. Were they measured on a likert-scale? Yes/no? each Question had the same responses as for all the questions in the survey? Where did the 31 total sum come from? Summing all responses? Were they in the same direction?

Line 170: this was an online survey. Anyone could have answered if respondents received the link. How can you be sure it was only filled by Vet Med students?

Line 192: why was this cutoff point selected? Why not, for example, 75%? Or 80%?

Why were the responses converted from a likert scale into 0 and 1? This is a descriptive study and readers may want to see the actual scores? Such conversion will limit the data and affect the comparison between groups of subjects.

Results

Some variables in table 1 could be collapsed to make more sense of the presented data.

Figures are not clear and not of any added value.

Line 219: some of the percentages presented in this section are scary. Especially for Vet Med students who should be more informed about the disease. Knowledge questions should be properly presented in the methods.

Some of the knowledge points in table 2 are still controversial. How this was decided to be true or false is still under scientific discussion.

Table 3: a p value above 0.05 is not statistically significant but could be used to decide on variables for regression models. Please modify the table notes accordingly.

Also, university name and region may be collinear with each others. Having them in the same model will introduce noise and redundancy.

Also, sample size calculations are conducted for a KAP study where a prevalence is presented. Conducting a logistic regression may not be a good idea especially with the number within each cell of the table. Collapsing the categories within each variable is needed to be able to conduct a logistic regression analysis.

Accordingly, the results may not be sound and scientific and the conclusions are limited by such.

Reviewer #3: The authors have gathered enough data and subjected the data appropriate statistical analysis and the conclusions drawn are appropriate. The manuscript was well written and i recommend that it be published as submitted.

6. PLOS authors have the option to publish the peer review history of their article (what does this mean?). If published, this will include your full peer review and any attached files.

Reviewer #1: **Yes: **Ado Danazumi Geidam

Reviewer #2: **Yes: **khalid A Kheirallah

Reviewer #3: **Yes: **Prof. J. D. Amin

---

## [Author Response · Author response to Decision Letter 0]

9 Dec 2020

A file responding to reviewers and editors comments have been uploaded

---

## [Decision Letter · Decision Letter 1]

4 Jan 2021

PONE-D-20-28765R1

SARS-CoV-2 (COVID-19 Pandemic) in Nigeria: Multi-institutional Survey of Knowledge, Practices and Perception Amongst Undergraduate Veterinary Medical Students

PLOS ONE

Dear Dr. Adebowale,

Thank you for submitting your manuscript to PLOS ONE. After careful consideration, we feel that it has merit but does not fully meet PLOS ONE’s publication criteria as it currently stands. Therefore, we invite you to submit a revised version of the manuscript that addresses the points raised during the review process.

There are still concerns on methodological clarity of the manuscript and the external validity of the results. At present the methods is not sufficiently described to support independent replication of the study. Please clarify and include a statement in the manuscript to confirm that the online survey was based on the Checklist for Reporting Results of Internet E-Surveys (CHERRIES), and guidelines for good practice in the conduct and reporting of online research.

Eysenbach G. Improving the quality of Web surveys:the Checklist for Reporting Results of Internet E-Surveys (CHERRIES) [published correction appears in doi:10.2196/jmir.2042] J Med Internet Res. 2004;6(3):e34. 10.2196/jmir.6.3 e34

It will be good if the authors, based on their local knowledge, provide an estimate of the total number of all the veterinary medical students (DV1 to DV5) in the study population (the 10 enlisted universities) that were eligible to participate in the study. This way the readers can make their independent judgment on the external validity of the results beyond the calculated sample size and the response rate. The last sentence of the limitation section seems tenuous. It is ambiguous to claim that self-selection and respondent biases or issue of generalizability  can be eliminated by having a response rate above the minimum estimated sample size. Using only limited number of participants in each school constitute a threat to the external validity of the results that is worth further discussion in the limitation section of the manuscript.

If possible, the authors should also consider to include the flow chart of the participants recruitment process to clearly indicate how many participants from each school click on the survey link (how many did not attempt any question), how many started the survey (how many started but did not complete) and how many finally completed the survey. Please take careful attention to address these and the other concerns of the reviewers.   

We look forward to receiving your revised manuscript.

Kind regards,

Adewale L. Oyeyemi, Ph.D

Academic Editor

PLOS ONE

Reviewers' comments:

Reviewer's Responses to Questions

**Comments to the Author**

1. If the authors have adequately addressed your comments raised in a previous round of review and you feel that this manuscript is now acceptable for publication, you may indicate that here to bypass the “Comments to the Author” section, enter your conflict of interest statement in the “Confidential to Editor” section, and submit your "Accept" recommendation.

Reviewer #1: All comments have been addressed

Reviewer #2: (No Response)

2. Is the manuscript technically sound, and do the data support the conclusions?

Reviewer #1: Yes

Reviewer #2: Yes

3. Has the statistical analysis been performed appropriately and rigorously? 

Reviewer #1: Yes

Reviewer #2: No

4. Have the authors made all data underlying the findings in their manuscript fully available?

Reviewer #1: Yes

Reviewer #2: Yes

5. Is the manuscript presented in an intelligible fashion and written in standard English?

Reviewer #1: Yes

Reviewer #2: Yes

6. Review Comments to the Author

Reviewer #1: 131-133----"The total sample size was divided equally among the veterinary schools (43 students per university). We purposively recruited a minimum of nine at each level (DVM 1-DVM 5) across the participating universities" – this is more like a Quota sampling; So the statement in 135 -136 “Purposive and convenience chain referral sampling technique was performed for student recruitment” needs to be rephrase

Reviewer #2: Statistical methods used to calculate the sample size related to KAP study and descriptive statistics using percentages and means for KAP. It does not allow comparisons and regression analyses. This was suggested initially and was not fully addressed. Using only a limited number of participants from each school does not mean you can compare by year and school, for example. An online survey is suited to recruit a large number of participants as this is a method to overcome other validity issues. As well, you will never be sure that those who filled the survey are actually those intended to.

7. PLOS authors have the option to publish the peer review history of their article (what does this mean?). If published, this will include your full peer review and any attached files.

Reviewer #1: **Yes: **Ado Danazumi Geidam

Reviewer #2: **Yes: **Khalid A. Kheirallah

---

## [Author Response · Author response to Decision Letter 1]

3 Feb 2021

RESPONSES TO EDITOR’S AND REVIEWERS’ COMMENTS (REVISION 2)

The authors are grateful for the comments and suggestions provided by the reviewers, which we found helpful and improved the quality of manuscript. Thank you Sirs

Below are our responses to the comments, accordingly.

S/N Location/lines revised Reviewer’s comment Remarks(Authors’ responses)

Plos One_ Editor’s queries/comments 

1. There are still concerns on methodological clarity of the manuscript and the external validity of the results. At present the methods is not sufficiently described to support independent replication of the study. Please clarify and include a statement in the manuscript to confirm that the online survey was based on the Checklist for Reporting Results of Internet E-Surveys (CHERRIES), and guidelines for good practice in the conduct and reporting of online research.

Response:

Thank you and we have clarified and confirmed in the manuscript that the online survey was based on the checklist for reporting results of internet E – surveys (CHERRIES), and guidelines for good practice in the conduct and reporting of online research (201 -203). We also ensured it was stated in the manuscript the participation rate of more than 100.0% (lines 254 -255), that a minimum of 9 students were then conveniently sampled from each level (141-143), the completion of the online survey took about 8 minutes and designed to ensure duplicate entries was avoided by preventing users with the same IP address access to the survey twice access to the survey twice (197 -199), ethical approval and informed consent described, development and pre-testing of the questionnaire was mentioned, survey administration was also provided in detail including the type of e-survey, survey context, that it was voluntary for participant, incentives offered has been included (lines 157 – 158).

2. It will be good if the authors, based on their local knowledge, provide an estimate of the total number of all the veterinary medical students (DV1 to DV5) in the study population (the 10 enlisted universities) that were eligible to participate in the study. This way the readers can make their independent judgment on the external validity of the results beyond the calculated sample size and the response rate.

Response: 

The enlisted number of universities in the study were 11. The total number of all the veterinary medical students eligible to participate in the study was 3724 (lines 131 -132). 

3. The last sentence of the limitation section seems tenuous. It is ambiguous to claim that self-selection and respondent biases or issue of generalizability can be eliminated by having a response rate above the minimum estimated sample size. Using only limited number of participants in each school constitute a threat to the external validity of the results that is worth further discussion in the limitation section of the manuscript.

Response: 

Thank you Sir. The authors quite agree that the last statement seems ambiguous and that generalizability cannot be eliminated by having a response rate above the minimum estimated sample size, coupled with other several limitations associated with the online survey e.g. the non-probabilistic sampling technique used. 

1. We have further expatiated the limitation of getting low participation from some schools, which was attributed to inaccessibility to the internet or inability to afford money to purchase data and insufficient power supplies (common complaints by the students) 2. We recognise for the online survey, random sampling was also impossible so we resulted to convenience sampling and voluntary participation, which also may have introduced selection bias. Due to challenges peculiar to the country as well as this study, we are therefore cautious with generalizing the result or outcomes of study to both veterinary student populations and across contexts. 

“Some limitations of the approach utilized in the study were identified. The introduction of enrolment and reporting biases may have resulted from the online survey making it non-representative. For the study, randomization was impossible due to the national lockdown, which could have possibly eliminated some of the biases. The non- probabilistic sampling approach, which is convenience and voluntary may have contributed to in the uneven distribution (coverage and non-participation) of student respondents from the different universities investigated. Furthermore, the poor internet accessibility or connectivity in the country (which may have varied from one institution location to another) and lack of funds to purchase data as complained by some students may have contributed to the lack of access to online questionnaire and participation. We are therefore cautious in generalizing the sample findings to the whole veterinary student populations in the country due to these limitations”.

4. If possible, the authors should also consider to include the flow chart of the participants’ recruitment process to clearly indicate how many participants from each school click on the survey link (how many did not attempt any question), how many started the survey (how many started but did not complete) and how many finally completed the survey.

Response: The data we have i.e. The total number of students that filled and completed the questionnaire, which is = 437. I have tried to include a flow chart as requested. The authors however, think since all participants accessed the link and completed filling questionnaire a flow chart may not be necessary.

FUNAAB = 204 75 (17.2)

UI = 454 73 (16.7)

UNILORIN = 250 26 (5.9)

UNIABUJA = 392 71 (16.2)

UNIJOS = 190 27 (6.2)

UNIMAID = 500 18 (4.1)

UDUS = 500 47 (10.8)

ABU= 400 14 (3.2)

MOUAU = 250 8 (1.8)

UAM = 194 60 (13.7)

UNN = 390 18 (4.1)

Response to Reviewer 1

Materials and Methods

 Lines 131-133 “The total sample size was divided equally among the veterinary schools (43 students per university). We purposively recruited a minimum of nine at each level (DVM 1-DVM 5) across the participating universities" – this is more like a Quota sampling; So the statement in 135 -136 “Purposive and convenience chain referral sampling technique was performed for student recruitment” needs to be rephrase Thank you so much Sir. The authors quite agree with this suggestion. We have therefore rephrased to read “

Reviewer 2

Materials and Methods

 Reviewer #2: Statistical methods used to calculate the sample size related to KAP study and descriptive statistics using percentages and means for KAP. It does not allow comparisons and regression analyses. This was suggested initially and was not fully addressed. Using only a limited number of participants from each school does not mean you can compare by year and school, for example. An online survey is suited to recruit a large number of participants as this is a method to overcome other validity issues. As well, you will never be sure that those who filled the survey are actually those intended to. Yes, we agree that the formula we used is for survey and not a comparative study as we did not set out to compare in the first place, as observed in case control, cohort and random clinical trials.

However, by virtue of the outcome variables that were categorized, we decided to look at possible independent variables that could explain the observed categorization (into good or poor).

---

## [Decision Letter · Decision Letter 2]

22 Feb 2021

SARS-CoV-2 (COVID-19 Pandemic) in Nigeria: Multi-institutional Survey of Knowledge, Practices and Perception Amongst Undergraduate Veterinary Medical Students

PONE-D-20-28765R2

Dear Dr. Adebowale,

We’re pleased to inform you that your manuscript has been judged scientifically suitable for publication and will be formally accepted for publication once it meets all outstanding technical requirements.

Kind regards,

Adewale L. Oyeyemi, Ph.D

Academic Editor

PLOS ONE

Reviewers' comments:

Reviewer's Responses to Questions

**Comments to the Author**

1. If the authors have adequately addressed your comments raised in a previous round of review and you feel that this manuscript is now acceptable for publication, you may indicate that here to bypass the “Comments to the Author” section, enter your conflict of interest statement in the “Confidential to Editor” section, and submit your "Accept" recommendation.

Reviewer #1: All comments have been addressed

2. Is the manuscript technically sound, and do the data support the conclusions?

Reviewer #1: Yes

3. Has the statistical analysis been performed appropriately and rigorously? 

Reviewer #1: Yes

4. Have the authors made all data underlying the findings in their manuscript fully available?

Reviewer #1: Yes

5. Is the manuscript presented in an intelligible fashion and written in standard English?

Reviewer #1: Yes

6. Review Comments to the Author

Reviewer #1: (No Response)

7. PLOS authors have the option to publish the peer review history of their article (what does this mean?). If published, this will include your full peer review and any attached files.

Reviewer #1: **Yes: **Ado Danazumi Geidam

---

## [Editor Report · Acceptance letter]

1 Mar 2021

PONE-D-20-28765R2 

SARS-CoV-2 (COVID-19 Pandemic) in Nigeria: Multi-institutional Survey of Knowledge, Practices and Perception Amongst Undergraduate Veterinary Medical Students 

Dear Dr. Adebowale:

I'm pleased to inform you that your manuscript has been deemed suitable for publication in PLOS ONE. Congratulations! Your manuscript is now with our production department. 

Kind regards, 

on behalf of

Dr. Adewale L. Oyeyemi 

Academic Editor

PLOS ONE